# Dental Epithelial Stem Cells as a Source for Mammary Gland Regeneration and Milk Producing Cells In Vivo

**DOI:** 10.3390/cells8101302

**Published:** 2019-10-22

**Authors:** Lucia Jimenez-Rojo, Pierfrancesco Pagella, Hidemitsu Harada, Thimios A. Mitsiadis

**Affiliations:** 1Department of Orofacial Development and Regeneration, Institute of Oral Biology, Centre for Dental Medicine, University of Zurich, 8032 Zurich, Switzerland; lucia@jimenez-rojo.com (L.J.-R.); pierfrancesco.pagella@zzm.uzh.ch (P.P.); 2Division of Developmental Biology and Regenerative Medicine, Department of Anatomy, Iwate Medical University Yahaba, Morioka 020-0023, Japan; hideha@iwate-med.ac.jp

**Keywords:** dental epithelial stem cells, mammary gland, tooth, tissue regeneration, rodent incisor, plasticity

## Abstract

The continuous growth of rodent incisors is ensured by clusters of mesenchymal and epithelial stem cells that are located at the posterior part of these teeth. Genetic lineage tracing studies have shown that dental epithelial stem cells (DESCs) are able to generate all epithelial cell populations within incisors during homeostasis. However, it remains unclear whether these cells have the ability to adopt alternative fates in response to extrinsic factors. Here, we have studied the plasticity of DESCs in the context of mammary gland regeneration. Transplantation of DESCs together with mammary epithelial cells into the mammary stroma resulted in the formation of chimeric ductal epithelial structures in which DESCs adopted all the possible mammary fates including milk-producing alveolar cells. In addition, when transplanted without mammary epithelial cells, DESCs developed branching rudiments and cysts. These in vivo findings demonstrate that when outside their niche, DESCs redirect their fates according to their new microenvironment and thus can contribute to the regeneration of non-dental tissues.

## 1. Introduction

Ectodermal appendages, such as hair follicles, teeth, salivary and mammary glands, are highly specialized structures that develop through continuous molecular crosstalk between epithelium and mesenchyme. These organs exhibit morphological and regulatory similarities during the earliest stages of development [1,2]. Molecular fine-tuning at more advanced developmental stages defines organ specificity and function. Variations in the expression of regulatory molecules can lead to conversion of a specific developing organ into another having completely different functions [3]. Classical tissue recombination experiments have highlighted the importance of these regulatory signals during ectodermal organ development and also revealed that the mesenchyme contains the morphogenetic potential [4,5]. Therefore, epithelial cells can be redirected and adopt alternative fates under the influences of tissue-specific mesenchymal signals.

All ectodermal organs contain stem cell populations that can generate various cell types, thus ensuring their homeostasis and regeneration throughout life [1,6]. Stem cell plasticity is enhanced during tissue regeneration, a process that requires, instantaneously, a large number of cells for the replacement of damaged or lost cells [6,7]. Various epithelial stem cell populations have been identified recently by genetic lineage tracing experiments in organs, such as hairs, mammary glands and teeth [6,8,9]. In the continuously erupting rodent incisors, multipotent dental epithelial stem cells (DESCs) that express stem cell markers such as Gli1 [8], Bmi1 [10] and Sox2 [9] are localized at their posterior end, the so-called cervical loop epithelium. Although these stem cells can generate all dental epithelial lineages, their capacity to form tissues of non-dental origin has not been assessed yet.

In vivo cell transplantation assays have been commonly used to identify epithelial stem cells and assess their plasticity during tissue regeneration [6,11,12]. Mammary gland regeneration is one of the most commonly used reconstitution assays, where mammary gland-derived epithelial fragments or disaggregated epithelial cells are transplanted into an epithelium-free mammary mesenchyme (also called fat pad) for de novo generation of functional ductal epithelial structures [11,13]. Expansion of the bilayered mammary epithelium occurs under the influence of hormones and is sustained by adult epithelial stem cells [14,15]. Classic chimeric recombination models demonstrated that the epithelial component is highly malleable, and that cell fate and tissue function are strongly influenced by the stromal component of the mammary gland. These studies using the mammary reconstitution assay indeed have shown that also neuronal, testicular, bone marrow and cancer cells mixed together with MECs can be reprogrammed and integrate into the epithelial ductal outgrowths [16,17,18]. However, these non-mammary epithelial cells never showed the ability to grow in mammary stroma without the support of MECs [12] and they were not able to generate ducts [16,17].

Here we aimed to demonstrate for the first time the capacity of DESCs to give rise to non-dental cell lineages that contribute to the formation of epithelial structures in organs other than teeth.

## 2. Materials and Methods

### 2.1. Cell Isolation and Lentiviral Infection

DESCs were isolated from the cervical loop of incisors of GFP-expressing Slc:dddy mice at postnatal day 3. These cells were cloned twice and cultured in collagen-coated plates in Dulbecco’s Modified Eagle Medium (DMEM)/F12 containing penicillin/streptomycin (1%), B27 (1X), epidermal growth factor (EGF) (200 ng/mL) and basic fibroblast growth factor (bFGF) (25 ng/mL). Mouse mammary epithelial cells (MECs) were isolated as described previously [19]. Briefly, mammary glands from eight week-old female mice were manually minced and enzymatically digested. After purification steps to get rid of the non-epithelial mammary tissues, epithelial organoids were obtained. A treatment with trypsin and DNase followed by filtration through a 40 μm cell strainer was performed in order to get single epithelial cells. The resulting MECs were plated in collagen-coated plates and infected with a lentivirus expressing DsRed prior to the in vivo mammary transplantation assay.

### 2.2. Animals and Surgical Procedures

All mice were maintained and handled according to the Swiss Animal Welfare Law (Animal License: 11/2014). Twenty-one day-old Rag1-/- (B6;129S7- Rag1<tm1Mom>/J(#002216); MGI: J1934) immunocompromised mice were used as hosts for the mammary transplantation experiments. Briefly, mammary fat pad containing endogenous epithelium (from the lymph node to the nipple area) was removed from the fat pads of the fourth mammary glands. Ten microlitres of cells (see Table 1 for cell amounts) in matrigel/phosphate buffered saline (PBS) (1:8) were injected in the remaining fat6 pad of mammary glands using a Hamilton syringe (with a 22-gauze needle). A total of 22 fat pads were inoculated. The mammary fat pads from virgin (never mated) host mice were dissected eight weeks after cell injection. At this time point, some of the mice were mated and plug checked in order to analyse them at pregnancy day 16 (P16). Whole mount pictures of the freshly dissected mammary glands were taken using a stereoscope. Afterwards, mammary glands were fixed for 30 min in paraformaldehyde 4% at room temperature and processed for paraffin embedding.

### 2.3. Immunofluorescence and Immunohistochemistry

Paraffin sections (4 μm) were stained with antibodies against GFP (A11122, Invitrogen, Basel, Switzerland; sc-9996, Santa Cruz Biotechnology, Heidelberg, Germany; ab6673, Abcam, Cambridge, UK), β-casein (sc-30042,), keratin 14 (PRB-155P, Biolegend, San Diego, CA, USA), oestrogen receptor-alpha (sc-542, Santa Cruz Biotechnology, Heidelberg, Germany), alpha-smooth muscle actin (MS-113-P0, Thermo Scientific, Basel, Switzerland; ab5694, Abcam, Cambridge, UK), fibronectin (ab2413, Abcam, Cambridge, UK), cytokeratin 8 (ab53280, Abcam, Cambridge, UK), amelogenin (ab59705, Abcam, Cambridge, UK). Briefly, sections were deparaffinized, rehydrated, and antigen retrieval was perform using 10 mM trisodium citrate buffer (pH 6.0) followed by permeabilization with 0.1% Triton X-100 and blocking with bovine serum albumin 1%. Sections were incubated with primary antibody overnight at 4 °C. For immunofluorescence, secondary antibodies (A11029; A10037; A21206; A10042, Invitrogen, Basel, Switzerland) were added for 45 min at RT and sections were incubated with 4′,6-diamidino-2-phenylindole (DAPI). Immunohistochemistry was performed for the detection of GFP-expressing cells during late pregnancy. Sections were, in this case, treated with 3% H_2_O_2_ for 10 min before antigen retrieval and a biotinylated secondary antibody was used. Then, sections were incubated with ABC reagent (VECTASTAIN Elite ABC Kit, Vector Laboratories, Peterborough, UK) for 30 min and staining was developed with SIGMAFAST™ 3,3′-diaminobenzidine (DAB) (D4168, Sigma-Aldrich, Buchs, Switzerlands).

For cell immunofluorescence staining, cells were fixed for 10 min in paraformaldehyde 4%, blocked with 1% BSA followed by incubation with primary antibodies against Sox2 (ab59776, Abcam, Cambridge, UK), Bmi1 (ab38295, Abcam, Cambridge, UK), oestrogen receptor-alpha (sc-542, Santa Cruz Biotechnology, Heidelberg, Germany), alpha-smooth muscle actin (MS-113-P0, Thermo Scientific, Basel, Switzerland; ab5694, Abcam, Cambridge, UK), fibronectin (ab2413, Abcam, Cambridge, UK), Islet1 (ab109517, Abcam, Cambridge, UK), E-cadherin (610182, BD Biosciences, Allschwil, Switzerland; AF748, R&D Systems, Minneapolis, MN, USA), keratin 14 (Biolegend PRB-155P), Notch1 [20], Notch2 [20], Tbx1 (Santa Cruz sc-17877), and vimentin (M0725, DAKO, Baar, Switzerland) 1 h at RT. Thereafter, cells were incubated 1 h at RT with secondary antibodies (A10042; A10037, Invitrogen, Basel, Switzerland) and finally stained with DAPI. Samples were mounted with DABCO mounting medium (Sigma-Aldrich, Buchs, Switzerlands) or ProLong Diamond Antifade mounting medium (Thermo Fisher Scientific, Basel, Switzerland) and imaged with a Leica DM6000 FS or a Leica SP8 CLSM microscope.

### 2.4. RNA Extraction and Real Time Quantitative Polymerase Chain Reaction (RT-qPCRs)

RNA was extracted using the RNeasy minikit (Qiagen, Hombrechtikon, Switzerland) and precipitated in ethanol. Then, retrotranscription was performed using iScript™ cDNA Synthesis Kit (Bio-Rad, Basel, Switzerland). Resulting cDNA was diluted 1:10 (to use approximately 5 ng per reaction) before performing real time quantitative PCR (RT-qPCR) using Eco™ Real-Time PCR System (Illumina, Zurich, Switzerland). Each target gene was internally normalized to the housekeeping gene 36B4. All primer sequences are available upon request.

## 3. Results

We used the mammary gland reconstitution in vivo assay to assess the plasticity and ability of DESCs to generate mammary epithelial cells after transplantation into the fat pads (Figure 1A). DESCs expressing both epithelial and stem cell markers (Figure 1B, Appendix A) were obtained from the cervical loop of GFP incisors via a cloning assay. These DESCs were mixed with primary mammary epithelial cells (MECs) and injected into epithelium-free mammary fat pads of immunocompromised RAG1 (recombination activating gene 1) -/- mice (Figure 1A). The fate of cells derived from these two cell populations was tracked by green fluorescent protein (GFP) expression for DESCs and lentivirus-induced DsRed (*Discosoma* sp. red) fluorescent protein expression for MECs (Figure 1A). Injection of MECs alone was used as positive control (Appendix A).

DESCs and MECs cells formed chimeric ductal structures composed by GFP-positive DESCs-derived cells and DsRed-positive MECs in mammary glands analysed eight weeks post-transplantation (Figure 1C–I) and pregnancy day 16.5 (Figure 1J,K).

To analyse in detail the distribution of transplanted DESCs within the developing chimeric ducts we first performed double immunofluorescence staining against GFP and keratin14 (Krt14), which is a marker for basal/myoepithelial cells in adult mammary gland [21] (Figure 2). GFP-positive cells were observed in both Krt14-positive myoepithelial and Krt14-negative luminal compartments (Figure 2C–K). DESCs-derived cells accounted for approximately 20% of the cells composing the epithelial compartment of the chimeric mammary ducts (Appendix A). Mammary luminal epithelium is complex and composed by various cell populations [22,23], grouped in two main subsets named ductal and alveolar cells. Ductal cells are lining the epithelial ducts and among them, oestrogen receptor alpha (ERα) expressing cells are responsible for the activation of the paracrine signalling that is essential for mammary epithelium elongation upon exposure to pubertal oestrogens [24]. On the other hand, alveolar cells constitute the milk-secreting alveolar units that arise during late pregnancy. Double immunofluorescence against GFP and ERα in the chimeric epithelium revealed that GFP-positive cells can give rise to both ERα-positive and ERα-negative luminal cells (Figure 2L–N). The ability of GFP-positive cells to give rise to luminal cells was further confirmed via double immunofluorescent staining against GFP and keratin 8 (Appendix A). Importantly, immunofluorescent and immunohistochemical analysis showed that GFP-positive DESCs could adopt a fully functional phenotype of β-casein-positive, milk-producing alveolar cells (Figure 2O–R, Appendix A).

We then wished to know whether DESCs possess the plasticity and reprogramming competence to regenerate ducts in absence of mammary epithelium. For this purpose, we performed the previously described mammary reconstitution assays in absence of MECs by injecting only DESCs into the fat pads (Figure 3A). Whole-mount fluorescence analysis revealed the formation of GFP-expressing small branched epithelial structures (Figure 3B, Appendix A). Immunofluorescence against GFP and detailed histological analysis showed that these outgrowths were exclusively composed by DESC-derived cells (Figure 3E) and surrounded by a dense fibrotic tissue (Figure 3C,D). The morphology of the cells composing the rudimentary ducts was variable, from flattened to columnar shapes (Figure 3D). To further analyse the composition of these ducts we assessed the distribution of Krt14 and ER-α. Immunofluorescent staining showed that these structures were composed by both Krt8-expressing luminal and Krt14-positive myoepithelial cells (Figure 3F,G, Appendix A). ER-α expressing cells were also detected in the epithelium of these ducts, although ER-α expression was significantly lower than that observed in the fully developed chimeric mammary epithelial outgrowths (Figure 3H). Similarly, we detected expression of βcasein within some DESC-derived ducts (Figure 3I), indicating that these cells have the capacity to initiate differentiation towards milk-producing, alveolar cells. At the same time, these cells show expression of amelogenin, a typical marker of ameloblast differentiation [25] (Figure 3J), thus indicating at least a partial memory of the tissue of origin. In some cases, the generated structures were dilated and adopted a cystic appearance (Figure 3K–M), characterized by flattened epithelial cells forming a monolayer (Figure 3L,M). These cystic structures were observed in more than 80% of the glands in which only DESCs were engrafted.

In association with DESCs transplantation we frequently observed the formation of dense fibrotic tissue. Fibrosis was detected in a subset of mammary fat pads transplanted with mixed MECs/DESCs, and in all fat pads inoculated with DESCs alone (Figure 4A,B). The non-epithelial tissue surrounding the ducts was composed by a very dense network of collagen fibres, as shown by Masson’s trichrome staining (Figure 4C,D). To assess the origin of the fibrotic tissue, we performed double immunofluorescent stainings for GFP and smooth muscle actin (α-SMA), a marker for fibrotic-tissue associated myofibroblasts (α-SMA). Double GFP- and α-SMA- positive DESCs-derived cells were detected in non-epithelial components of the mammary gland, i.e., in the fibrotic tissue surrounding the ducts (and cysts (Figure 4E,F). The tissue surrounding ducts and cysts showed presence of fibronectin (Figure 4G), a fundamental component of mammary stroma as well as fibrotic tissue. Double GFP- and fibronectin-positive DESC-derived cells could be observed in the fibrotic tissue surrounding the ducts (Figure 4H). These results suggest that DESCs can adopt a mesenchymal fate and generate myofibroblasts.

## 4. Discussion

In this study, we aimed to assess the plasticity of dental epithelial stem cells (DESCs) and their capacity to contribute to the regeneration of non-dental organs, namely mammary glands. By transplanting DESCs together with mammary epithelial cells (MECs) in the mammary gland fat pad we demonstrated that DESCs can be fully reprogrammed to a mammary epithelial phenotype. In this context DESCs gave rise to all the different cell types that compose the mammary epithelium, including milk-producing cells. We showed also that DESCs injected alone can form a small ductal system. DESCs are thus capable to initiate without inputs from mammary epithelium a ductal branching morphogenesis, process characteristic of many developing organs, such as mammary and salivary glands, and lungs, but not representative of odontogenesis [1]. This feature is, to our knowledge, a prerogative of DESCs. Indeed, previous studies using the mammary reconstitution assay have shown that neuronal, testicular, bone marrow and cancer cells mixed together with MECs can be reprogrammed and integrate into the epithelial ductal outgrowths [16,17,18]. However, non-mammary epithelial cells never showed the ability to grow in mammary stroma without the support of MECs [12] and they were not able to generate ducts [16,17]. Our results thus provide the first evidence that mammary branching structures can be formed in fat pads in absence of mammary epithelial inputs. Tooth and mammary gland development display several molecular and morphological similarities in their initial stages. Both organs belong to the class of ectodermal appendages and develop as a result of continuous and reciprocal interactions between epithelium and mesenchyme. The initial stages of mammary and tooth development largely depend on the same molecular cues, mostly associated to transforming growth factor β (Tgf-β), fibroblast growth factor (Fgf) and Wnt signalling pathways [26,27]. Mutations in genes coding for Wnt ligands lead to similar defects in tooth and mammary development, as they cause tooth agenesis [28] and mammary glands aplasia [29]. At later stages, however, dental and mammary development significantly diverge [1], as these organs undergo very distinct morphogenesis. These could be due to a combination of differences in the responsiveness to molecular cues and in the stromal microenvironment. A plethora of signalling pathways is involved in mammary gland morphogenesis. Among these, the FGF pathway is required from the first phases of mammary gland development and throughout ductal elongation and branching [30,31]. The various FGF ligands exert different effects on their target tissues, and within mammary gland development they control distinct aspects of ducts formation. FGF-10 regulates branch initiation, which depends on directional epithelial migration. By contrast, FGF-2 controls ductal elongation, which in turn requires cell proliferation and epithelial expansion [32]. Similarly, FGF signalling is an important driver of tooth development. FGF-10 and FGF-3 are not involved in tooth initiation, but they regulate tooth morphogenesis by stimulating epithelial cell proliferation in the cervical loops [33] and are important for the maintenance of DESCs in the mouse incisor [34,35,36]. DESCs are, thus, responsive to FGF-10 and FGF-3, while no evidence supports any effect of FGF-2 onto these cells. Our results show that DESCs transplanted alone could drive the formation of smaller ducts compared to the transplantation of a mixture of DESCs and MECs. Mammary stroma-derived FGF-10 might, thus, induce branch initiation upon transplantation of FGF10-responsive DESCs. At the same time, DESCs might not be able to respond to FGF-2-mediated proliferative stimuli and, thus, fail to give rise to full-size ducts.

Another mechanism underlying the suboptimal morphogenesis displayed by DESCs might lie in the activation of ERα expression. ERα-expressing cells are responsible for the activation of the paracrine signalling that is essential for mammary epithelium elongation, and ERα knock-out mice display severely impaired ductal elongation [24,37,38,39]. ERα is expressed at low levels in cultured DESCs (Appendix A), and it was expressed at low levels in the small ducts formed by DESCs alone, making them poorly responsive to the hormonal stimuli. Our results suggest that MECs provide molecular signals that induce high expression of ERα in DESCs, making the latter responsive to hormones and, thus, capable of contributing to a full branching morphogenesis.

Upon transplantation of solely DESCs, and in a subset of transplantations of mixed MECs/DESCs, the regeneration of rudimentary ducts was associated with the generation of dense fibrotic tissue. During wound healing processes, early inflammation is followed by proliferation and differentiation of fibroblasts into α-SMA-expressing myofibroblasts that actively secrete extracellular matrix (ECM) prior to tissue remodelling. When this process is dysregulated, myofibroblasts can remain active and produce an excess of ECM that results in a pathological condition known as fibrosis, which impairs the normal tissue repair or regeneration process [40,41]. The ECM is a major regulator of mammary epithelial architecture and function, as it provides not only physical support but also essential molecular cues that guide cellular fate and function of the gland [27,42]. Stromal ECM proteins are key cues for the development of the mammary arboreal structure. It was shown for example that collagen I fibres orientation in the mammary fat pad patterns mammary branch orientation [42]. Overall, proper tissue architecture and stiffness of the ECM are necessary components of normal development, differentiation and function within the mammary gland [43]. Indeed, our results showed that some of the DESCs might be able to undergo epithelial-to-mesenchymal transition (EMT), differentiate into α-SMA-expressing myofibroblasts and thereby contribute to fibrotic tissue formation. This process could be induced by stimulation of DESCs with TGFβ1, which is present within the mammary stroma in gradients. Lower concentrations of TGFβ1 are permissive for outgrowth and branching, while higher levels restrain ducts formation via Wnt5a [44,45]. EMT in combination with ECM deposition and degradation is necessary for proper mammary epithelial invasion [31], but a misbalancing of the system can lead to pathological fibrosis. It is known that TGFβ1 inhibits proliferation of ERα-expressing MECs in vivo [46] and at the same time it stimulates EMT as well as ECM deposition [47]. It may be that signals from mammary epithelium are essential for reversing the activation of myofibroblasts and for preventing excessive DESCs EMT. In this scenario, MECs could be fundamental to control TGFβ1-mediated signalling, promoting ERα-expression in DESCs, preventing their EMT and allowing full ductal outgrowth. We observed that DESCs express already prior to transplantation mesenchyme-associated genes, such as *Vim* (vimentin) and *Acta2* (smooth muscle actin) (Appendix A). This might indicate an innate tendency of DESCs to adopt multiple epithelial fates and potentially mesenchymal fates.

The experiments have been performed with DESCs isolated from incisors extracted from young mouse pups (PN3). It is well established that stem cell potency changes throughout development and decreases by ageing [48], but no thorough study ever characterized age-dependent changes in the DESCs niche of rodent incisors. Only a recent article proposed that the incisors’ DESCs niche might reach a steady, homeostatic state at approximately eight weeks of age [49]. This steady state is defined however on the basis of proliferation dynamics, and not DESCs plasticity or differentiation potential. Thus, to date there is no evidence indicating that DESCs would present age-dependent alterations in their plasticity [9,49] and no studies ever investigated the potential of DESCs to adopt other than dental cell fates in vivo.

Taken together, these results clearly demonstrate that DESCs can generate all mammary epithelial cell lineages in de novo regenerated mammary ducts, indicating their plasticity and multi-lineage differentiation potential (Figure 5). Moreover, we demonstrate that DESCs are to date the only cell type capable of initiating mammary-specific branching morphogenesis in the absence of any mammary epithelium. This is the first ever in vivo evidence showing that DESCs could be redirected by a definite microenvironment to adopt other than dental cell fates, displaying an exceptional degree of plasticity. These results thus clearly demonstrate the potential of DESCs for the regeneration of tissues and organs of non-dental origin by adopting both epithelial and mesenchymal cell fates.

## Figures and Tables

**Figure 1 cells-08-01302-f001:**
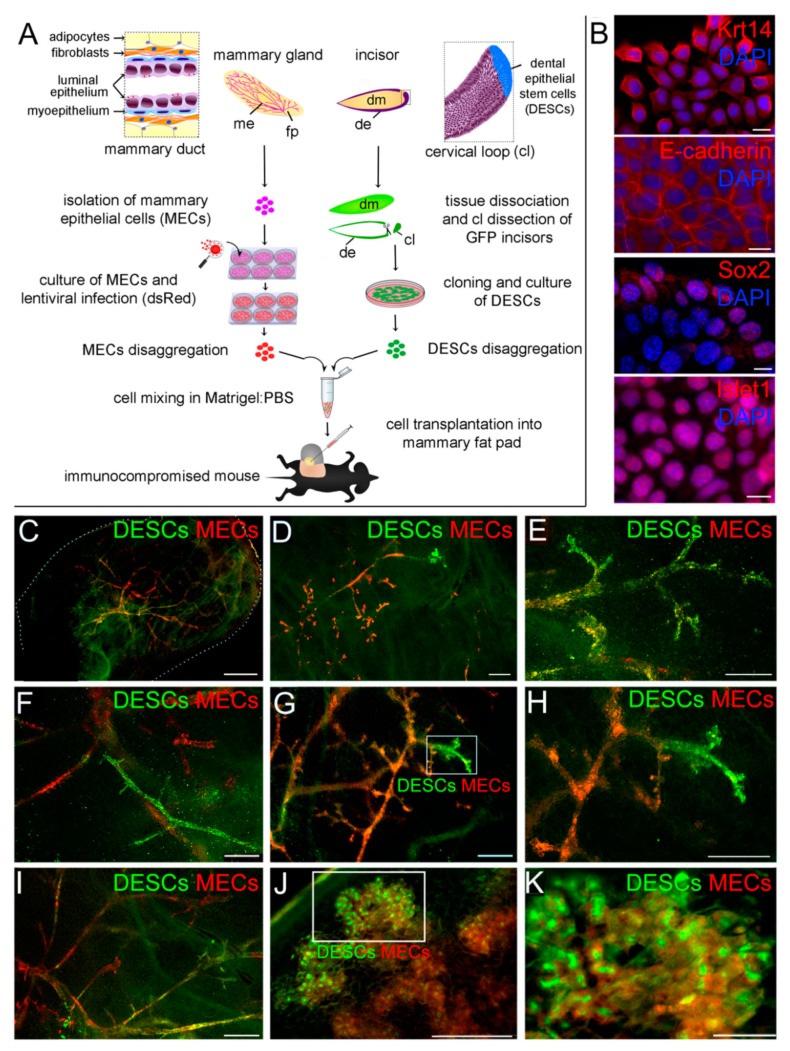
Injection of DESCs and MECs into mammary fat pads results in the formation of a chimeric ductal epithelium. (**A**) GFP-DESCs and DsRed-mammary epithelial cells (MECs) were mixed and injected into the mammary fat pads of immunocompromised mice. (**B**) Before injecting them into the mammary microenvironment, DESCs expressed epithelial markers such as keratin 14 (Krt14) and E-cadherin (E-cad); the dental epithelial stem cell marker Sox2 and the incisor epithelium marker Islet1. (**C**–**K**) Whole mount fluorescent imaging of epithelial outgrowths from virgin (**C**–**H**) and pregnancy day 16.5 (**J**,**K**) chimeric mammary glands. Boxes in (**G**) and (**J**) represent the areas of high magnifications in (**H**) and (**K**), respectively. Scale bars: 25 μm (**B**); 2 mm (**C**,**J**); 400 μm (**D**–**I**,**K**). Abbreviations: cl, cervical loop; de, dental epithelium; DESCs, dental epithelial stem cells; dm, dental mesenchyme; fp, fat pad; me, mammary epithelium; MECs, mammary epithelial cells.

**Figure 2 cells-08-01302-f002:**
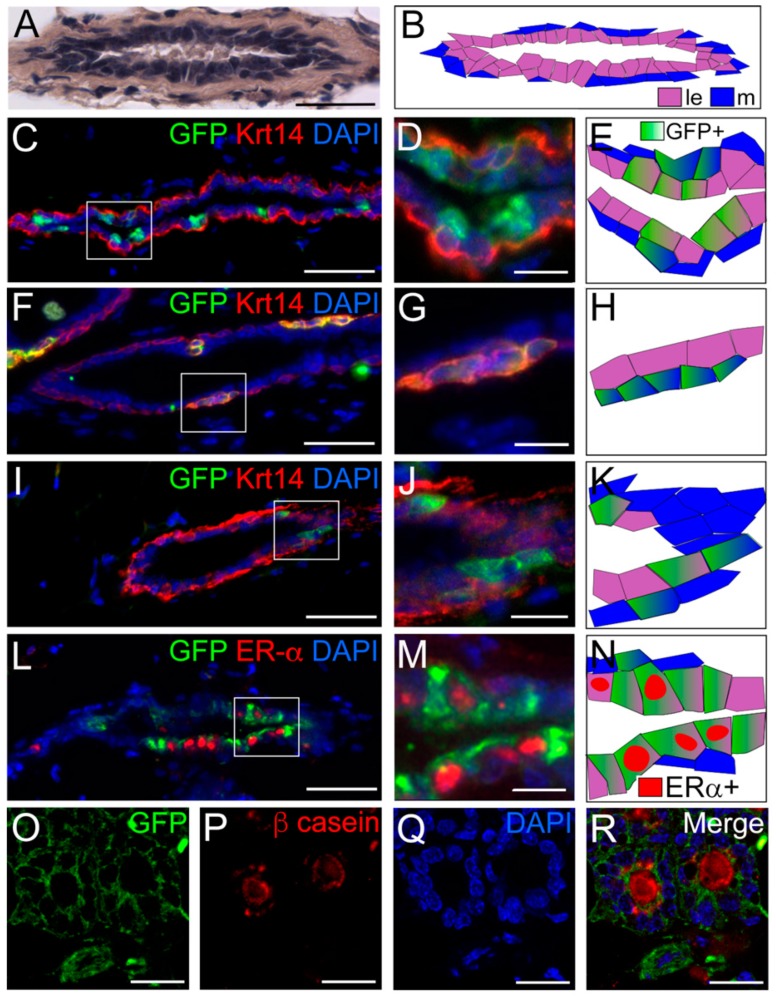
DESCs give rise to different cell lineages of mammary epithelium. (**A**,**B**) Hematoxylin-eosin staining of the chimeric ducts (**A**) and schematic representation (**B**). (**C**–**N**) Double immunofluorescence against Krt14 and GFP (**C**,**D**,**F**,**G**,**I**,**J**) and against oestrogen receptor alpha (ERα) and GFP (**L**,**M**), and schematic representations of the various types of alveolar cells (**E**,**H**,**K**,**N**) showing the integration of GFP positive cells (DESC-derived) within the different compartments of the chimeric mammary ducts. Boxes in C,F,I,L represent high magnifications shown in D,G,J and M. (**O**–**R**) Immunofluorescent staining against GFP and β-casein. (**O**–**Q**) Single channels; (**R**) merged image. Scale bars: 50 μm (**A**); 40 μm (**C**,**F**,**I**,**L**); 10 μm (**D**,**G**,**J**,**M**); 20 μm (**O**–**R**). Abbreviation: ld, lipid droplet.

**Figure 3 cells-08-01302-f003:**
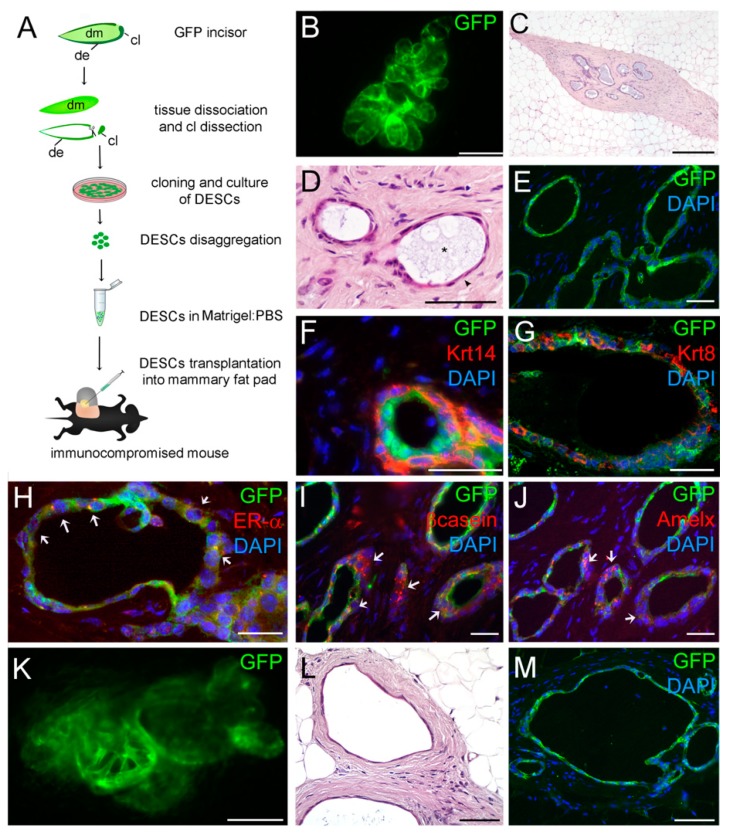
Injection of DESCs alone in mammary fat pads. (**A**) GFP^+^ DESCs injected alone in mammary fat pads managed to form epithelial branched structures (green colour). (**C**,**D**) Hematoxylin-eosin staining showing DESCs-originated ducts. Notice the presence of secretions within the ducts (asterisk in D). (**E**) Immunofluorescent staining showing that ducts are originated exclusively by GFP^+^ DESCs (green colour). (**F**,**G**) Immunofluorescent staining showing Krt14-expressing myoepithelial cells (**F**) and Krt8-expressing luminal cells (**G**) originated from DESCs. (**H**–**J**) Immunofluorescent staining against GFP (green colour) and ERα (**H**), β-casein (**I**), or amelogenin (**J**; red colour). Arrowheads indicate double-positive cells for each combination of staining. (**K**–**M**) Fluorescent imaging (**K**), H and E staining (**L**), and immunofluorescent staining against GFP (**M**) showing cyst-like structures originated from DESCs. Scale bars: 400 μm (**B**,**K**), 100 μm (**C**–**E**,**H**–**J**,**L**,**M**); 50 μm (**F**,**G**). Abbreviations: le, luminal epithelium; m, myoepithelium.

**Figure 4 cells-08-01302-f004:**
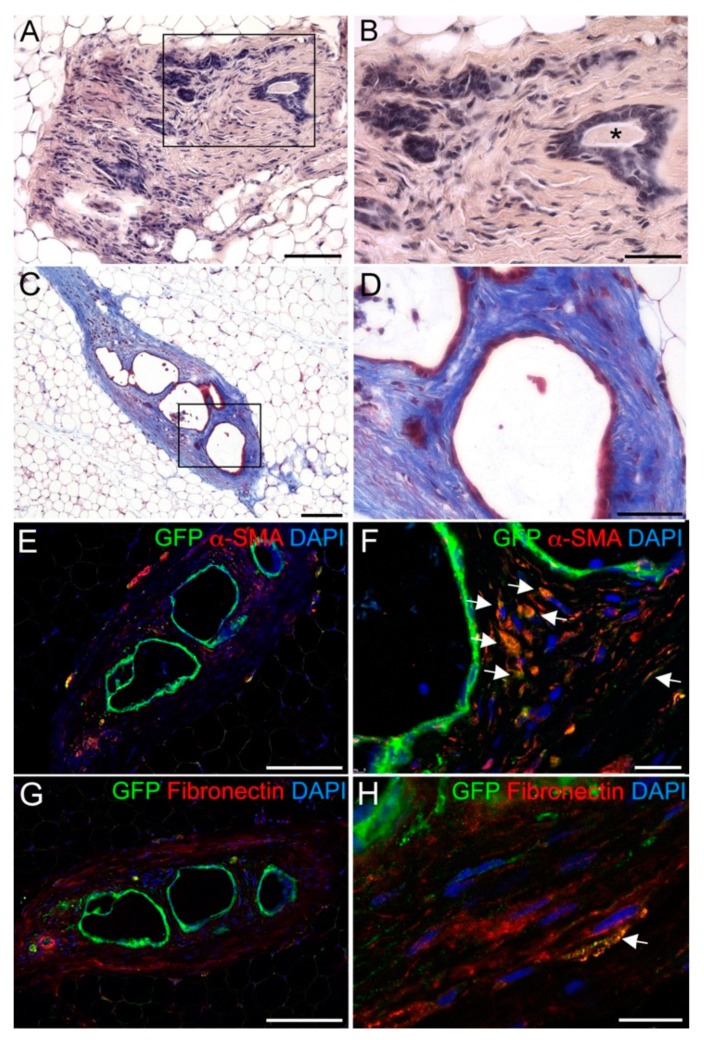
DESCs integrate in epithelial rudiments and potentially surrounding stroma. (**A**,**B**) Hematoxylin-eosin staining showing the formation of dense fibrotic tissue upon transplantation of DESCs. Black rectangular box in A indicates the region shown in B. Notice the presence of secretions within the ducts (asterisk in B). (**C**,**D**) Masson’s trichrome staining showing the composition of the fibrotic tissue surrounding the ducts. Black rectangular box in C indicates the region shown in D. (**E**,**F**) Double immunofluorescent staining against GFP and αSMA. White arrowheads indicate some of the double GFP^+^/α-SMA^+^ cells. (**G**,**H**) Double immunofluorescent staining against GFP and Fibronectin. White arrowheads indicate double GFP^+^/Fibronectin^+^ cells. Scale bars: 100 μm (**B**); 200 μm (**A**,**C**,**E**,**G**); 20 μm (**F**,**H**).

**Figure 5 cells-08-01302-f005:**
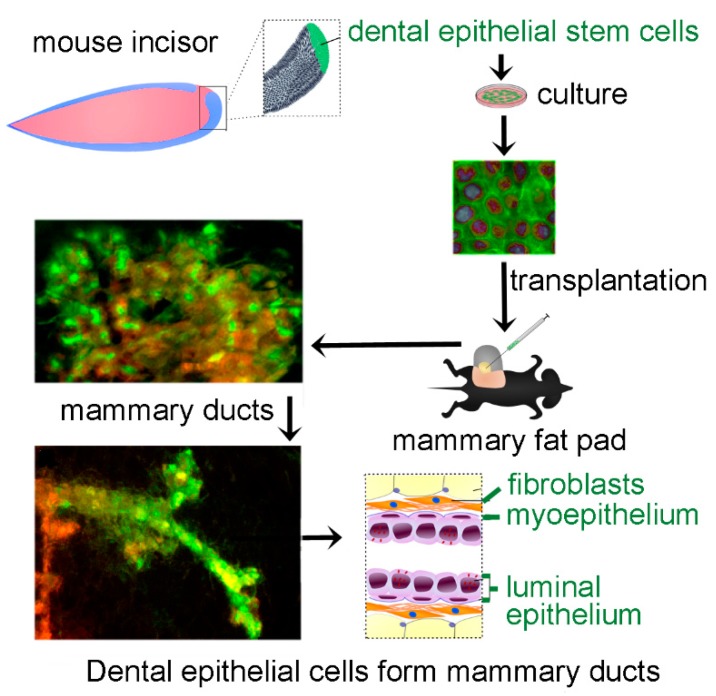
Schematic representation recapitulating the experimental approach and the main results showing the plasticity of dental epithelial stem cells and their potential to adopt mammary epithelial cell fates.

**Table 1 cells-08-01302-t001:** Summary of the obtained outgrowths after injection of DESCs, MECs, and combinations of both cell types into mammary fat pads. The last row (asterisks) corresponds to the fat pads that were analysed at pregnancy day 16.

No. of Dental Epithelial Stem Cells (DESCs)	No. of Mammary Epithelial Cells (MECs)	No. of Inoculated Fat Pads	No. of Intakes	No. of Outgrowths
50,000	50,000	7	6	6
100,000	50,000	4	3	3
100,000	0	6	6	6
50,000 *	50,000 *	5 *	5 *	5 *

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
