# Peer review of "Dental Epithelial Stem Cells as a Source for Mammary Gland Regeneration and Milk Producing Cells In Vivo"

_cells, 2019, doi:10.3390/cells8101302_

Round 1

Reviewer 1 Report

The study of Jimenez-Rojo and colleagues provides new insights into the regenerative capacity of organ-dependent stem cells. The authors used dental epithelial stem cells to regenerate the mammary gland, meaning that stem cells are able to shape their fate in a microenvironment-dependent manner, and to regenerate into non-dental tissues, specifically mammary tissue when injected in the mammary gland. This is a striking and novel finding that well suits to be published at Cells. However, additional data is required in order to define the capacity of DESCs being regenerated into mammary tissue.

The part of EMT must be eliminated in the manuscript unless authors perform rigorous assays and clearly demonstrate it in vitro and in vivo,as it is right now it is not acceptable. The mouse strain used for DESCs and MECs isolation is not specified. This must be mentioned in order to justify the use of immunocompromised mice as receptors for transplantation experiments. Same findings should be demonstrated in an immunocompetent model K8 must be used in all the regenerated glands to show luminal cells Fig3c must show a zoom out image with the H&E staining of ducts and branching. It is not clear that this are ducts. I would like to recommend a mammary reconstitution with MaSC should be used as a Control in each condition. Using MEC+MaSC (in figure 2), and MaSC alone (in figure 3) as control injections is necessary in this case. Next, GFP and b-casein co-expression by immunofluorescence should be assessed in figure 2 the same way as done in figure 3. Moreover, an image of the whole mammary gland stained with Carmine should be shown, which would allow the evaluation of duct formation and branching. Also, it is unclear which mice are being used to obtain the mammary glands used to study the secretion of b-casein. Are these the P16 mice mentioned in line 87 (Materials and methods)? A control gland of each condition needs to be shown, as mentioned in the previous point. In line 191, structures with a cystic morphology are defined. Since these only appear in DESC-derived mammary glands, they should be quantified in order to judge their relevance. A high amount of fibrotic tissue is observed surrounding ducts in DESC-derived mammary glands, but this doesn’t mean that fibroblasts composing of this tissue are actually in a fully active state. Since the staining of SMA is weaker than expected in an activated tissue, cells composing of this tissue should be better characterized. Masson's Trichrome staining of collagen fibers should be performed in order to support their observations. Images in figure 5 show cells in a high confluence, which is not ideal when studying EMT states. EMT experiments should be repeated using subconfluent cultures, IF stainings should be quantified, and other EMT markers need to be assessed by RTqPCR or WB in order to quantify whether E-cadherin decrease is enough to consider EMT state.

Author Response

We thank the reviewer for the thorough reviewing of the manuscript and her/his feedback. Reviewer's comments are reported in regular-black characters, our replies in red-italics.

Review 1

The study of Jimenez-Rojo and colleagues provides new insights into the regenerative capacity of organ-dependent stem cells. The authors used dental epithelial stem cells to regenerate the mammary gland, meaning that stem cells are able to shape their fate in a microenvironment-dependent manner, and to regenerate into non-dental tissues, specifically mammary tissue when injected in the mammary gland. This is a striking and novel finding that well suits to be published at Cells.

We thank the reviewer for appreciating the great relevance of our work.

However, additional data is required in order to define the capacity of DESCs being regenerated into mammary tissue.The part of EMT must be eliminated in the manuscript unless authors perform rigorous assays and clearly demonstrate it in vitro and in vivo,as it is right now it is not acceptable.

Following the input from the reviewers, we decided to remove the experiments concerning TGF-b1-dependent induction of EMT in DESCs (Fig. 5). We strengthened the evidence supporting in vivo EMT and the contribution of DESCs to the formation of fibrotic tissue by providing new immunostaining as well as more thorough histological characterization.

The mouse strain used for DESCs and MECs isolation is not specified. This must be mentioned in order to justify the use of immunocompromised mice as receptors for transplantation experiments. Same findings should be demonstrated in an immunocompetent model.

DESCs were obtained from the outbred strain Slc:ddy, while MECs were isolated from C57BL/6 mice. Given that the cells had different backgrounds, we decided to use immunocompromised mice to avoid possible rejection problems. Although the role and effects of the immune system in regenerative processes is of importance, the goal of this article is to assess the plasticity and potential of DESCs to give rise to non-dental tissues. The study of the effects of the immune responses on this process is thus beyond the scopes of this work.

K8 must be used in all the regenerated glands to show luminal cells.

We included images of co-staining against GFP and Keratin 8 in Figure 3 and as supplementary figures (Figure S4).

The presence of GFP positive cells in the luminal compartment is supported by the location of GFP-expressing cells facing the ductal lumen of the regenerated epithelium. Those cells present the typical luminal morphology and do not express the basal/myoepithelial marker K14. In addition, we show that some of those GFP-positive cells express ERa, which is always expressed by a subset of luminal cells but never expressed by the basal/myoepithelial cells (Tornillo and Smalley., 2015: https://www.ncbi.nlm.nih.gov/pmc/articles/PMC4595529/#CR18). The fact that ERa is specific for luminal cell lineages and that we found ERa-expressing GFP cells, implies that some GFP cells are able to acquire luminal fates.

Fig3c must show a zoom out image with the H&E staining of ducts and branching. It is not clear that this are ducts.

A picture obtained at lower magnification has been added. The epithelial structures developed by DESCs injected alone do not fully recapitulate the normal morphology of mammary epithelium. The resulting ducts can initiate a branching morphogenetic program in a similar way as it occurs at mammary embryonic development. However, they do not present secondary and tertiary branching as normal mammary epithelium does.

 I would like to recommend a mammary reconstitution with MaSC should be used as a Control in each condition. Using MEC+MaSC (in figure 2), and MaSC alone (in figure 3) as control injections is necessary in this case.

We did not add to our experiments the transplantation of MEC+MaSC together since we assume that MaSCs are also present among the injected MEC population. The aim our work was not to compare between the ability of DESCs and MaSCs/MECs to reconstitute mammary epithelium.

We did perform, as positive control, a set of experiments in which only MECs were injected. We include now in Figure S2 an image obtained by the whole-mount analysis of the mammary epithelium obtained after grafting only DsRed MECs in cleared fat pads.

Next, GFP and b-casein co-expression by immunofluorescence should be assessed in figure 2 the same way as done in figure 3.

We provide a double immunofluorescent staining in Figure 2 (O-R), where co-staining of GFP and bcasein is visible.

Usually lactating glands show non-specific auto-fluorescence that makes difficult to interpret the results obtained by double-immunofluorescence assays. Due to this technical limitation, we  analysed the expression of GFP and bcasein by immunohistochemistry in the same alveolar structure by using two consecutive sections (5 um each). We chose to magnify an alveolus that is fully formed by GFP positive cells in one section to show in the consecutive section that bcasein is expressed by the cells from that alveoli and the milk protein is also present in the lumen of the alveolus. The fact that some of the GFP positive cells in the mammary alveoli from chimeric glands show the presence of lipid droplets in the cytoplasm strongly supports that they have been differentiated into functional, milk-producing mammary alveolar cells, even in the absence of a direct co-staining for b-casein.

Moreover, an image of the whole mammary gland stained with Carmine should be shown, which would allow the evaluation of duct formation and branching.

Carmine alum is a classical technique that stains mammary ducts and requires clearing the adipose tissue with organic solvents to further analyze the whole gland using a stereoscope. This technique has been broadly used in order to analyze the generated mammary ducts in transplantation experiments when the epithelial cells do not present any kind of marker. In our experiments, both MECs and DESCs express fluorochromes that allow evaluation of formation of epithelial structures without the need of staining with the classical Carmin Alum staining. Thus, we performed a whole-mount fluorescent analysis of freshly harvested fat pads to detect GFP and DsRed expression and the resulting epithelial structures without the need of using any additional staining procedure (Fig 1). This approach thus substitutes the classic Carmine alum staining. Carmine alum has indeed the great inconvenient of being most of the times incompatible with further analysis of the sample by immunohistochemical or immunofluorescence-based techniques due to the treatment of the gland with highly aggressive solvents.

Also, it is unclear which mice are being used to obtain the mammary glands used to study the secretion of b-casein. Are these the P16 mice mentioned in line 87 (Materials and methods)?

Yes, in order to analyse the contribution of GFP-expressing cells to milk-producing glands, we analysed some of the transplanted glands at late pregnancy stages (P16). This has been clarified in the manuscript.

A control gland of each condition needs to be shown, as mentioned in the previous point.

We included a whole mount showing the ducts formed by MECs injected alone as Figure S2. We also included another positive control, i.e. b-casein IHC on a P16 mammary epithelium obtained by injecting MECs without DESCs, in Figure S5. We also

In line 191, structures with a cystic morphology are defined. Since these only appear in DESC-derived mammary glands, they should be quantified in order to judge their relevance.

We observed the formation of cystic structures in 5 out of 6 mammary glands in which DESCs alone were engrafted. This information has been added to the text.

A high amount of fibrotic tissue is observed surrounding ducts in DESC-derived mammary glands, but this doesn’t mean that fibroblasts composing of this tissue are actually in a fully active state. Since the staining of SMA is weaker than expected in an activated tissue, cells composing of this tissue should be better characterized. Masson's Trichrome staining of collagen fibers should be performed in order to support their observations.

We provided new immunofluorescent staining against aSMA and GFP, imaged via high resolution confocal microscopy. These images show abundant expression of aSMA in cells located around the ducts, and many of these cells show expression of GFP, indicating that they are derived from GFP-positive DESCs. We provided also Masson’s Trichrome staining, to highlight the presence of abundant collagen fibers in the fibrotic tissue surrounding the ducts of DESCs-derived mammary glands.

Images in figure 5 show cells in a high confluence, which is not ideal when studying EMT states. EMT experiments should be repeated using subconfluent cultures, IF stainings should be quantified, and other EMT markers need to be assessed by RTqPCR or WB in order to quantify whether E-cadherin decrease is enough to consider EMT state

Following the feedback from the reviewers, figure 5 has been removed from the manuscript

Reviewer 2 Report

In this work Jimenez-Rojo et al. investigated on the plasticity of dental epithelial stem cells (DESCs) in vivo. In particular the authors tested the plasticity of DESCs (GFP+) in presence or absence of GFP- mammary epithelial cells (MECs) into mammary fat pads.  The combination of the GFP+ DESCs and GFP- MECs resulted in chimeric ductal epithelium with GFP+ cells differentiated into Krt14-positive myoepithelial, ductal and alveolar cells. In the absence of MMECS instead DESCs developed only branching rudiments and cysts. The authors also described the presence of GFP+ non-epithelial cells, possibly myofibroblasts, indicating that DESCs are able to undergo EMT. In support of this hypothesis, DESCs treated in vitro with tgfb1 expressed a marker of mesenchymal cells.   

The manuscript unfolds logically and the data are very interesting. However, to support the conclusions, some histological quantification is needed for some experiments relative to Fig 2 and 3. Concerning the EMT claims I feel more convincing evidences are required. Overall though I think that the manuscript would be very interesting even in absence of the EMT transition data (fig.3 and 4) but with more quantification relative to Fig. 2 and 3 and a better characterization of the DESCs in culture conditions.

Here below some points that should be taken in consideration before publication:

About Fig. 1B, why these markers have been used? Sox2 is a transcription factor so should be in the nucleus. Additional lower magnification images showing cells in culture  conditions would be useful to characterize better the cell heterogeneity in cultured DECTs.

To exclude the possibility that some GFP positive fibroblasts are not present in these cultures the author should also check for some fibroblast markers (this is required to support the EMT-like observed in vivo in fig 4).

In Fig. 1 c-k would be useful to have dapi staining to visualize gfp and DsRed negative structures.

All the results are interesting but the is a general lack of quantification: in particular Figure 2,3,4,5. At the moment the only quantification report is on the text: “ER-α expressing cells were also detected in the epithelium of these ducts, although ER-α expression was significantly lower than that observed in the fully developed chimeric mammary epithelial outgrowths (Fig. 3H).”

The authors claim that GFP-positive DESCs (Fig. 2O-Q) could adopt a fully functional phenotype. If a co-staining with gfp and B-casein is not possible, a histological analysis with tissue slices from consecutive cutting would strongly support their claim.

The author also claims that DESCs integrated into surrounding stroma as ACTA2 positive myofibroblasts.  The white arrows in panels C and D in fig.4 indicate mainly cell fragments in which green and red signal do not colocalize in most of the cases. How the author explains this? More images of double positive cells should be provided. Even if only a small proportion of non-epithelial GFP+ cells will be positive for ACTA2, the authors might maintain their claims.

In term of mechanism, to prove that in vitro tgfb1 is able to start changing the identity of epithelial cell towards a mesenchymal phenotype, more evidences are needed. At the moment there is only one mesenchymal marker. More fibroblast markers are required.  Functional assay to test adhesion or migration might also be required to prove EMT.

In addition to prove that tgfb1 treated DESCs undergo EMT, the authors could inject them into mammary fat pads and check if the number of GFP+ myofibroblast increase in comparison with untreated DESCs.

Author Response

We thank the reviewer for the insightful comments and feedback. We here provide a point to point reply (red, italics) to the comments (black, regular).

Reviewer 2

In this work Jimenez-Rojo et al. investigated on the plasticity of dental epithelial stem cells (DESCs) in vivo. In particular the authors tested the plasticity of DESCs (GFP+) in presence or absence of GFP- mammary epithelial cells (MECs) into mammary fat pads.  The combination of the GFP+ DESCs and GFP- MECs resulted in chimeric ductal epithelium with GFP+ cells differentiated into Krt14-positive myoepithelial, ductal and alveolar cells. In the absence of MMECS instead DESCs developed only branching rudiments and cysts. The authors also described the presence of GFP+ non-epithelial cells, possibly myofibroblasts, indicating that DESCs are able to undergo EMT. In support of this hypothesis, DESCs treated in vitro with tgfb1 expressed a marker of mesenchymal cells.   

The manuscript unfolds logically and the data are very interesting. However, to support the conclusions, some histological quantification is needed for some experiments relative to Fig 2 and 3. Concerning the EMT claims I feel more convincing evidences are required. Overall though I think that the manuscript would be very interesting even in absence of the EMT transition data (fig.3 and 4) but with more quantification relative to Fig. 2 and 3 and a better characterization of the DESCs in culture conditions.

Here below some points that should be taken in consideration before publication:

About Fig. 1B, why these markers have been used?

We included a much more extensive characterization of mHAT DESCs in Figure S1.

Krt14 and E-Cadherin are markers of the dental epithelium:

Tabata et al. 1996 - https://www.sciencedirect.com/science/article/pii/S0003996996000878?via%3Dihub

Li et al. 2012 – https://www.sciencedirect.com/science/article/pii/S0012160612001479?via%3Dihub

Sox2 is an established DESCs marker:

e.g. Juuri et al. 2014 - https://www.sciencedirect.com/science/article/pii/S1534580712002390

Islet1 marks specifically incisor-derived epithelial cells:

Mitsiadis et al. 2003; https://dev.biologists.org/content/130/18/4451.long

Sox2 is a transcription factor so should be in the nucleus.

In mHAT14 cells Sox2 shows nuclear and perinuclear expression. Perinuclear expression of transcription factors such as Sox2 and Oct4 in stem and cancer cells has been previously described by others:

Li et al., 2016 - https://www.ncbi.nlm.nih.gov/pmc/articles/PMC5029659/#R24

van Schaijik et al., 2017 - https://www.ncbi.nlm.nih.gov/pubmed/29180509

Bhattacharya  et al., 2019 - https://www.ncbi.nlm.nih.gov/pubmed/30548157

We added other images of immunofluorescent staining showing clearer Sox2 nuclear localization in mHAT DESCs in Figure 1 and Figure S1.  

Additional lower magnification images showing cells in culture conditions would be useful to characterize better the cell heterogeneity in cultured DECTs. To exclude the possibility that some GFP positive fibroblasts are not present in these cultures the author should also check for some fibroblast markers (this is required to support the EMT-like observed in vivo in fig 4).

mHAT DESCs lines have been obtained from two-step cloning assays: all cells are thus derived from an epithelial clone. We nevertheless performed a thorough analysis by immunofluorescent staining. All cells express Islet 1 (Fig. S1), which is expressed solely in the dental epithelium and not in dental mesenchyme. All cells express on their surface E-Cadherin (Fig. S1 H-K; images of different focal planes are available, showing that all cells express on their surfaces E-Cadherin), marker of DESCs. Some cells however also express mesenchymal markers, such as vimentin and a-SMA, at the gene expression level and, to a lower extent, also at protein level. Albeit peculiar, this is in accordance with RNA sequencing analysis performed on freshly isolated mouse incisor dental epithelial stem/progenitor cells (unpublished data). This co-expression of epithelial-mesenchymal markers might indicate a tendency of DESCs to undergo EMT.

In Fig. 1 c-k would be useful to have dapi staining to visualize gfp and DsRed negative structures.

The images in Fig 1 (C-K) are not stained, we performed a whole-mount fluorescent analysis to detect GFP and DsRed expression before the fluorescence was lost during further stages (such as fixation and sample preparation for paraffin embedding). This procedure is performed when transplanted cells express a fluorescent marker and substitutes the classic Carmine alum staining that in a similar manner is used to detect the presence of mammary outgrowths after transplantation experiments. Carmine alum technique implicates treating the sample with aggressive solvents that are usually incompatible with further analysis of the sample by immunohistochemical or immunofluorescence techniques. Thus, the aim of the pictures in Figure 1 is to prove that chimeric outgrowths were generated from transplanted cells. Regions that do not express GFP and DsRed correspond to the cells from the fat pad of the donor, which are not the focus of the analysis.

All the results are interesting but the is a general lack of quantification: in particular Figure 2,3,4,5. At the moment the only quantification report is on the text: “ER-α expressing cells were also detected in the epithelium of these ducts, although ER-α expression was significantly lower than that observed in the fully developed chimeric mammary epithelial outgrowths (Fig. 3H).”

We included more quantitative information in the manuscript. We inserted a table reporting the numbers of experiments for each experimental condition (Table 1), the proportion of ducts vs cysts formed upon transplantation of DESCs in the fat pad (text), and the proportion of DESCs found within ducts formed upon transplantation of mixtures of DESCs / MECs (Fig. S3).

The authors claim that GFP-positive DESCs (Fig. 2O-Q) could adopt a fully functional phenotype. If a co-staining with gfp and B-casein is not possible, a histological analysis with tissue slices from consecutive cutting would strongly support their claim.

We provide double immunofluorescent staining in Figure 2O-R, where co-staining of GFP and bcasein is visible.

The immunohistochemistry presented previously in Figure 2, and now in Figure S5, correspond to two consecutive sections (5 um each). We chose to magnify an alveolus that is fully formed by GFP positive cells in one section to show, in the consecutive section, that b-casein is expressed by the cells from that alveoli and the milk protein is also present in the lumen of the alveolus. The fact that some of the GFP positive cells in the mammary alveoli from chimeric glands show the presence of lipid droplets in the cytoplasm strongly supports that they have been differentiated into functional, milk-producing mammary alveolar cells.

The author also claims that DESCs integrated into surrounding stroma as ACTA2 positive myofibroblasts.  The white arrows in panels C and D in fig.4 indicate mainly cell fragments in which green and red signal do not colocalize in most of the cases. How the author explains this? More images of double positive cells should be provided. Even if only a small proportion of non-epithelial GFP+ cells will be positive for ACTA2, the authors might maintain their claims.

We provided new immunofluorescent staining against aSMA and GFP, imaged via high resolution confocal microscopy. These images show abundant expression of aSMA in cells located around the ducts, and many of these cells show expression of GFP, indicating that they are derived from GFP-positive DESCs.

In term of mechanism, to prove that in vitro tgfb1 is able to start changing the identity of epithelial cell towards a mesenchymal phenotype, more evidences are needed. At the moment there is only one mesenchymal marker. More fibroblast markers are required.  Functional assay to test adhesion or migration might also be required to prove EMT. In addition to prove that tgfb1 treated DESCs undergo EMT, the authors could inject them into mammary fat pads and check if the number of GFP+ myofibroblast increase in comparison with untreated DESCs.

Following the input from the reviewers, we decided to remove the experiments concerning TGF-b1-dependent induction of EMT in DESCs (Fig. 5). The experiments proposed by the reviewer would be surely of interest to fully address whether and how DESCs consistently and robustly acquire and maintain a mesenchymal phenotype. This demonstration is however beyond the scopes and not necessary for this article, as the reviewer noted in her/his first comment.

Reviewer 3 Report

The manuscript by Jimenez-Rojo and colleagues convincingly shows that dental epithelial stem cell are able to generate chimeric mammary gland outgrowth when injected with mammary epithelial cells in mice cleared fat pads. In addition, when injected without mammary epithelial cells, dental epithelial cells can generate mammary gland rudiments in fat pads.

The manuscript is well written and the experiments realized strongly support the author’s conclusion.

I have a few minor comments :

1/ A few spelling mistakes need to be corrected

2/The Materials and Methods section mention a Table 1 where the amounts of mammary and dental epithelial cells injected should be mentioned. I can’t find this Table 1 in the downloaded manuscript. These informations would be important to assess what amount of dental epithelial cells compare to mammary epithelial cells needs to be injected to get these chimeric mammary glands formation.

3/The introduction section could maybe describe in greater details the fact that until this paper, no cell type had been shown to be able to generate mammary gland rudiments in absence of co-injected epithelial mammary cells. It would allow the reader a better understanding of the importance of the results shown before reading the discussion section.

4/The dental epithelial cells used are from very young pups (3 days old). Did the authors tried with older mice ? A discussion of that point could be of interest for the reader.

5/The major point of this manuscript is the ability of dental epithelial cells to form mammary rudiments without co-injected mammary cells in cleared fat pads. It seems then very important to ensure that the procedure used to clear the fat pads removed all endogenous epithelial mammary cells. Ideally the authors could show that not grafted cleared fat pads never gave mammary gland outgrowth.

Author Response

We thank the reviewer for the positive and constructive feedback.

We here reply point to point to the comments.

Review 3

 The manuscript by Jimenez-Rojo and colleagues convincingly shows that dental epithelial stem cell are able to generate chimeric mammary gland outgrowth when injected with mammary epithelial cells in mice cleared fat pads. In addition, when injected without mammary epithelial cells, dental epithelial cells can generate mammary gland rudiments in fat pads.

The manuscript is well written, and the experiments realized strongly support the author’s conclusion.

We thank the reviewer for the positive feedback.

I have a few minor comments:

1/ A few spelling mistakes need to be corrected

We thoroughly controlled the manuscript for spelling mistakes.

2/The Materials and Methods section mention a Table 1 where the amounts of mammary and dental epithelial cells injected should be mentioned. I can’t find this Table 1 in the downloaded manuscript. This information would be important to assess what amount of dental epithelial cells compare to mammary epithelial cells needs to be injected to get these chimeric mammary glands formation.

 We apologize for the mistake; we now added the Table.

3/The introduction section could maybe describe in greater details the fact that until this paper, no cell type had been shown to be able to generate mammary gland rudiments in absence of co-injected epithelial mammary cells. It would allow the reader a better understanding of the importance of the results shown before reading the discussion section.

 We added this important point in the Introduction.

4/The dental epithelial cells used are from very young pups (3 days old). Did the authors tried with older mice? A discussion of that point could be of interest for the reader.

 The point raised by the reviewer is of interest. The experiments described in this manuscript have been performed only with DESCs derived from P3 pups. No thorough studies assessing potential age-dependent differences in DESCs properties have been performed. Only a recent article from our group and collaborators (Sharir et al. 2019) proposed that the mouse incisor, and incisor DESCs, might reach a real homeostatic condition at 8 weeks of age. However, this is based on proliferation dynamics rather than plasticity. Both at early postnatal stages and in adult life, incisors’ DESCs can give rise to all dental epithelial lineages (Juuri et al. 2014; Sharir et al. 2019). No evidence exists concerning eventual age-dependent alterations in DESCs multipotency.

This point has now been addressed in the Discussion.

5/The major point of this manuscript is the ability of dental epithelial cells to form mammary rudiments without co-injected mammary cells in cleared fat pads. It seems then very important to ensure that the procedure used to clear the fat pads removed all endogenous epithelial mammary cells. Ideally the authors could show that not grafted cleared fat pads never gave mammary gland outgrowth.

All fat pads were thoughtfully analyzed by histological sections and hematoxylin-eosin staining did not reveal the presence of any epithelial structure other than the ducts and cysts that were surrounded by fibrotic tissue. In addition, in all pads injecting only with GFP-positive DESCs, all ducts / cysts expressed GFP, confirming that there was no contamination from a possible endogenous source.

Round 2

Reviewer 1 Report

Authors have addressed many of my concerns, but there are some issues still remaining. The manuscript should be ready for publication after few statement corrections or experimental reassessing.

Fig. 3 K8 staining is not clear for luminal cells, authors should show the localization of K8 cells showing only red images. If necessary, authors could isolate CD24+/CD29low luminal cells and measure K8 expression. The mammary gland reconstitution as shown in Fig.3c is also a concern. It is very fibrotic and do not show mammary ducts, authors should show whole mount gland (Carmine staining or fluorescent staining try to show ducts), if not possible please discuss better in the text that DESCs alone are not reconstituting the gland but creating few epithelial structures among fibrotic tissue. As it is right now, authors cannot say that these are branch structures. It should be mentioned that DESC are creating more fibrotic/mesenchymal tissue than epithelial tissue.

I recommend to tone down the tittle since DESCs by themselves do not regenerate the mammary gland.

Author Response

We thank the reviewer for the inputs. Here provide a point-to-point reply to the comments.

(Replies in red, italics).

------------

Authors have addressed many of my concerns, but there are some issues still remaining. The manuscript should be ready for publication after few statement corrections or experimental reassessing.

Fig. 3 K8 staining is not clear for luminal cells, authors should show the localization of K8 cells showing only red images. If necessary, authors could isolate CD24+/CD29low luminal cells and measure K8 expression.

We provided in Figure S4 single channel images showing the localization of K8-positive cells.

The mammary gland reconstitution as shown in Fig.3c is also a concern. It is very fibrotic and do not show mammary ducts, authors should show whole mount gland (Carmine staining or fluorescent staining try to show ducts), if not possible please discuss better in the text that DESCs alone are not reconstituting the gland but creating few epithelial structures among fibrotic tissue. As it is right now, authors cannot say that these are branch structures. It should be mentioned that DESC are creating more fibrotic/mesenchymal tissue than epithelial tissue.

We provided in Figure S6 higher resolution images showing branched structures formed by GFP-positive DESCs, highlighted by arrowheads. In the manuscript we show and discuss that DESCs indeed form short branches, albeit not a fully developed mammary epithelium. In the Discussion section, we also extensively discuss the possible molecular basis for the smaller growth of DESCs-derived branches.

I recommend to tone down the title since DESCs by themselves do not regenerate the mammary gland.

Following the input of the reviewer, we changed the title from:

“Dental epithelial stem cells as a source for mammary gland regeneration and milk producing cells formation in vivo”

to

 “Dental epithelial stem cells contribute to mammary gland regeneration and generate milk producing-cells in vivo”

Reviewer 2 Report

The authors revised their manuscript improving quality and quantity of data and providing some quantification. Overall I think these are very interesting data.

However I'm still not completely convinced that DESCs can differentiate into miofibroblasts. I think that more experiments would be required to fully claim this. Nevertheless I think that the data provided at the moment would be enough for publication with few text changes. In particular I would suggest to change the following text

row 299: "DESCs integrate in epithelial rudiments and potentially surrounding stroma"  

row 419: instead of "are able" I would put "might be able"

row 450: "adopting multiple epithelial fates and potentially mesenchymal fates"

Author Response

We thank the reviewer for the constructive inputs. We here reply to the comments raised (red, italics)

--------

The authors revised their manuscript improving quality and quantity of data and providing some quantification. Overall I think these are very interesting data.

We thank the reviewer for acknowledging the improvement of the manuscript.

However I'm still not completely convinced that DESCs can differentiate into miofibroblasts. I think that more experiments would be required to fully claim this. Nevertheless I think that the data provided at the moment would be enough for publication with few text changes. In particular I would suggest to change the following text

row 299: "DESCs integrate in epithelial rudiments and potentially surrounding stroma"  

row 419: instead of "are able" I would put "might be able"

row 450: "adopting multiple epithelial fates and potentially mesenchymal fates"

We can changed the sentences in accordance to the input from the reviewer.